# Antimicrobial Usage and Antimicrobial Resistance in Commensal *Escherichia coli* from Broiler Farms: A Farm-Level Analysis in West Java, Indonesia

**DOI:** 10.3390/antibiotics13121181

**Published:** 2024-12-05

**Authors:** Rianna Anwar Sani, Sunandar Sunandar, Annisa Rachmawati, Gian Pertela, Oli Susanti, Kanti Puji Rahayu, Puttik Allamanda, Imron Suandy, Nofita Nurbiyanti, Elvina J. Jahja, Budi Purwanto, Francisca C. Velkers, Tagrid Dinar, Jaap A. Wagenaar, David C. Speksnijder

**Affiliations:** 1Division of Infectious Diseases and Immunology, Faculty of Veterinary Medicine, Utrecht University, 3584 CL Utrecht, The Netherlands; r.anwarsani@uu.nl (R.A.S.); tagrideltahir@hotmail.com (T.D.); j.wagenaar@uu.nl (J.A.W.); 2Center for Indonesian Veterinary Analytical Studies (CIVAS), Bogor 16130, Indonesia; nando.nbx@gmail.com (S.S.); drhannisa95@gmail.com (A.R.); fietha@gmail.com (N.N.); 3Animal Health Department, PT Medion Farma Jaya, Bandung 40552, Indonesia; gian.pertela@medionindonesia.com (G.P.); elvina_j@medionindonesia.com (E.J.J.); 4National Quality Control Laboratory for Animal Product Testing and Certification (BPMSPH), Bogor 16161, Indonesia; iban_atar@yahoo.co.id (O.S.); kanti_pujirahayu@yahoo.co.id (K.P.R.); puttikallamanda@gmail.com (P.A.); imron_az@yahoo.com (I.S.); 5Technical Education & Consultation Department, PT Medion Ardhika Bhakti, Bandung 40223, Indonesia; budi_p@medionindonesia.com; 6Department of Population Health Sciences, Faculty of Veterinary Medicine, Utrecht University, 3584 CL Utrecht, The Netherlands; f.c.velkers@uu.nl; 7WHO Collaborating Center for Reference and Research on Campylobacter and Antimicrobial Resistance from a One-Health Perspective/WOAH Reference Laboratory for Campylobacteriosis, 3584 CL Utrecht, The Netherlands; 8Wageningen Bioveterinary Research, 8221 RA Lelystad, The Netherlands; 9University Farm Animal Clinic, 3481 LZ Harmelen, The Netherlands

**Keywords:** Indonesia, antimicrobial use, antimicrobial resistance, farm level, poultry, broilers, *E. coli*

## Abstract

**Background/Objectives:** Antimicrobial resistance (AMR) is a global public health threat, with antimicrobial use (AMU) in livestock recognized as a significant driver. This study examines farm-level AMU and AMR as well as the relationship between AMU and AMR on broiler farms in Indonesia. **Methods:** Data were collected from 19 farms in West Java between 2019 and 2021 to examine AMU in depth across four to five successive production cycles. The correlation between AMU and AMR in commensal *Escherichia coli* (*E. coli*) was investigated. AMU was recorded as treatment days per 30-day production cycle, and antimicrobial susceptibility was assessed using epidemiological cut-off (ECOFF) values to differentiate wildtype (WT) and non-wildtype (NWT) *E. coli*. **Results:** The average AMU was 12 treatment days per 30-day production cycle, with a wide range of 4 to 22 days. On average, *E. coli* isolates from each farm exhibited NWT phenotypes, reflecting AMR levels, for 6 out of 14 antimicrobials tested. This included notable levels for the highest priority critically important antimicrobials (HPCIAs) ciprofloxacin (93%) and nalidixic acid (64%). A significant correlation (Spearman ρ = 0.67, *p* < 0.05) was observed between the total farm-level AMU and the number of antimicrobials for which NWT *E. coli* isolates were found. However, no significant correlation was found between AMU and AMR for the five most frequently used antimicrobials, likely due to a high baseline prevalence of NWT *E. coli* isolates and relatively few independent observations. **Conclusions:** These findings highlight the urgent need to reduce AMU in general, specifically the use of (HP)CIAs, to mitigate AMR on Indonesian broiler farms.

## 1. Introduction

Antimicrobial resistance (AMR) is a serious public health threat associated with 4.95 million human deaths worldwide in 2019 [1]. Resistant bacteria can develop and spread in livestock due to antimicrobial use (AMU), creating a potential reservoir that can affect humans through direct contact, the food chain, or the environment [2,3,4,5,6,7]. As AMU has been associated with the increased occurrence and dissemination of AMR, reducing AMU in livestock could eventually result in a decrease in zoonotic AMR transmission [3,8,9,10]. This highlights the need to monitor both AMU and AMR in farm animals to facilitate risk management and policy making.

Indonesia acknowledges the public health threat of AMR and developed its first National Action Plan (NAP) in 2017 with a follow-up in 2022 [11,12]. Amongst other objectives, the NAP aims to stimulate prudent AMU and reduce the development of AMR from a one-health perspective by involving the livestock sector [11,12]. The specific focus is broiler production, which is a major part of the Indonesian livestock sector, as chicken meat accounts for 65% of the animal protein consumed by the Indonesian population [13]. Based on current trends, the broiler sector is projected to expand even further in the coming years [13]. Given the lack of professional veterinary oversight and widespread over-the-counter access to antimicrobials, regulating and monitoring their use in broilers remain challenging.

As one of the few countries in Southeast Asia, Indonesia implemented an ongoing surveillance system for AMR in broilers in 2019. Indonesia monitors AMU by collecting data through questionnaires sent to farmers and sales data from pharmaceutical companies. These provide valuable insights into AMU at the national and regional levels. Current surveillance data and previous studies have shown that AMU in the broiler sector is substantial, with a high proportion of preventive use (80%) and considerable use of highest priority critically important antimicrobials (HPCIAs), particularly fluoroquinolones [14,15]. AMU is projected to rise even further in the near future [9]. The World Health Organization (WHO) published a classification of antimicrobials based on their importance for human medicine and the potential risk of AMR to human health arising from antimicrobial use in non-human sectors [16]. In this list, the most important antimicrobials for human medicine are categorized as critically important antimicrobials (CIAs), with the highest priority ones categorized as HPCIAs [16]. Resistance to these antimicrobials therefore poses a serious risk to public health.

In addition to AMU data at the regional level, there is a need for specific AMU data at the farm level. Farm-level data can facilitate a better understanding of the correlation between AMU and AMR on the farm, which can facilitate risk assessment and management. Additionally, it can support the development of targeted interventions to foster prudent AMU in the livestock sector. While several studies have examined AMR and AMU in Indonesia’s poultry sector, there are limited quantitative and qualitative data on AMU at the farm level as well as the relationship between AMU and AMR at the farm level [15,17,18].

Therefore, the first objective of this study was to collect comprehensive data on AMU and AMR at the farm level in small- and medium-scale broiler farms in West Java, Indonesia. The second aim was to describe the association between the observed AMU and AMR on the farms studied.

## 2. Results

Data were collected from 78 production cycles across 16 small-scale and 3 medium-scale broiler farms in West Java, Indonesia. Of the 25 initially recruited farms, 6 small-scale farms dropped out of the study due to the cessation of farming activities or changes in ownership or management. The average capacity of each farm was 31,779 broilers (ranging from 4800 to 80,000). The mean number of broilers present in the study houses included during a production cycle was 8738 (ranging from 1715 to 25,000, SD: 6548). Housing systems varied within the farms with open, semi-closed, or closed housing systems (Table 1).

Open housing systems (also known as traditional houses) consist of an open system in the form of stilts, made of wood or bamboo [19], and are largely influenced by the external temperature and humidity conditions [20]. Closed housing systems have control of the temperature and humidity [19]. Semi-closed housing systems are open houses that have been modified so that there is some control over the temperature and/or humidity, but not to the same extent as closed housing systems [19].

Six farms had a prewritten standard (antimicrobial) treatment protocol in place. This standard treatment protocol is a suggested treatment scheme that is usually provided by pharmaceutical companies but can also be provided by an integrator or the farm owner. It typically includes advice to administer preventive antimicrobials to the flock on the first three days after the arrival of chicks on the farm and around vaccinations. Protocols differ depending on the provider. When asked who they consulted for veterinary advice, all but one farmer mentioned a technical support officer advising them on broiler health. One farmer (farm 18) consulted a veterinarian instead of a technical support officer. Technical support officers are often not veterinarians but rather graduates with a bachelor’s degree in animal science. The choice of antimicrobial was based on the standard treatment protocol, personal experience, or advice from technical support officers or veterinarians.

### 2.1. Overview of Antimicrobial Usage and Antimicrobial Susceptibility Testing

Within this study, 38 different veterinary medical products (VMPs) containing antimicrobials were used, of which 24 contained a combination of two active antimicrobial compounds (Figure 1).

One product was classified as an antimicrobial product by the farmers, but its specific content was unclear. This product was therefore excluded from the analysis. In total, eleven different classes of antimicrobials were used on our study farms (Table 2). The five most frequently applied antimicrobial classes on the farms in decreasing order of treatment days were fluoroquinolones, macrolides, tetracyclines, polymyxins, and penicillins.

The mean AMU per studied broiler house in count-based treatment frequency (TF_count-based_) was 0.39 (SD: 0.16), which means there was an average of 12 treatment days per production cycle (which has an average duration of 30 days). Farm 7 had the lowest AMU (average TF_count-based_ of 0.14, or 4 treatment days per production cycle), and farm 14 had the highest AMU (average TF_count-based_ of 0.73, or 22 treatment days per production cycle) (Figure 2; Table 2).

The AMU per farm showed consistency across successive production cycles on some farms (e.g., farms 3, 4, 14, and 15 (Appendix A)), while for others, there was considerable variation (e.g., farms 2, 6, and 7 (Appendix A)). This applied to both the quantities and the classes of antimicrobials used in the different cycles. Notably, on several farms that reported the availability of a standard treatment protocol (farms 1, 2, 5, 10, 11, and 15 (Table 1)), we also observed differences in the specific antimicrobials chosen and their frequency of use across different cycles.

Antimicrobial susceptibility data were collected in the fourth successive cycle on 17 farms and the fifth successive cycle on 2 farms (farms 4 and 15). The average time between the start of the first and the last successive cycles in our study period was 9 months (ranging from 8 to 20 months; SD 4.2 months). The tested isolates showed varying degrees of non-wildtype (NWT) phenotypes to both antimicrobials from classes that were used as well as classes that were not used on the farms within the study period (Table 2 and Table 3; Figure 2 and Figure 3). Antimicrobial classes for which NWT phenotypes were detected, despite not being used during any of the production cycles included in this study, included amphenicols (chloramphenicol), third- and fourth-generation cephalosporins (cefotaxime and ceftazidime), carbapenems (meropenem), and glycylcycline (tigecycline). The raw MIC data can be found in Appendix A.

Using the epidemiological cut-off (ECOFF values), the highest prevalence of NWT phenotypes on the study farms was observed for ciprofloxacin (93%), followed by ampicillin (88%), tetracycline (83%), sulfamethoxazole (75%), and trimethoprim (71%).

Additionally, when clinical breakpoints (CBPs) were used, the highest levels of resistance were found in decreasing order to ampicillin (88%), tetracycline (83%), sulfamethoxazole (74%), trimethoprim (71%), and nalidixic acid (57%) (Table 3; Figure 3).

An analysis of the 25 isolates collected per farm showed that, on average, the Escherichia coli (*E. coli*) isolates were NWT to 6 out of the 14 tested antimicrobials (SD: 1.16). Furthermore, when assessed using CBPs, the isolates displayed resistance to an average of 5 out of 14 antimicrobials (SD: 1.29).

Among the HPCIAs, the quinolones ciprofloxacin and nalidixic acid exhibited a median MIC value higher than the ECOFF (Figure 3). If CBPs were used, only the HPCIA quinolone nalidixic acid showed a median MIC higher than the CBP.

### 2.2. Correlation Analysis

At the clustered level, we found a Spearman correlation of 0.67 (*p* < 0.05; CI 0.25–0.89) between the average AMU per farm and the average number of antimicrobials for which NWT *E. coli* phenotypes were found per farm using ECOFF breakpoints (Figure 4).

Using CBP, we found a slightly higher Spearman correlation of 0.70 (*p* < 0.05; CI 0.31–0.90) between the average AMU per farm and the average number of antimicrobials for which the *E. coli* isolates were resistant per farm (Figure 5). Both results indicate a significant correlation between the total AMU per farm and the number of antimicrobials to which NWT *E. coli* isolates (ECOFF) or clinical resistance (CBP) were found.

For the five most frequently used antimicrobial classes (fluoroquinolones, macrolides, tetracyclines, polymyxins, and penicillins), we analyzed the odds ratio (OR) of changes in the proportion of NWT phenotypes or clinical resistance for tested antimicrobials within the same class when the number of treatment days of this specific antimicrobial class was increased by one treatment day. In all cases, the OR we calculated was not significant (*p* > 0.05 and all CIs containing 1.00).

## 3. Discussion

This study links on-farm AMU data to antimicrobial susceptibility patterns in *E. coli* isolates on broiler farms in Indonesia. Every farm used antimicrobials at least once per studied production cycle, but AMU varied widely between farms. Similarly, the prevalence of NWT phenotypes varied considerably between farms, with relatively high proportions of NWT phenotypes found for ampicillin, ciprofloxacin, and tetracycline. On some farms, NWT *E. coli* isolates were found when tested for antimicrobials belonging to classes that were not used in the past four to five production cycles within the study period.

A noticeable correlation was found when AMU and AMR were examined. The average AMU (defined as TF_count-based_ calculated as the average use over the monitored production cycles) showed a significant correlation with the average number of antimicrobials to which the *E. coli* isolates were found to be NWT per farm (ρ = 0.67; *p*-value 0.0016). This suggests that the average proportion of NWT phenotypes increases with an increase in average AMU per production cycle. However, caution is warranted in interpreting this correlation. Notably, the presence of NWT phenotypes was detected for four tested antimicrobial classes, although no antimicrobials belonging to these antimicrobial classes were used in any of the production cycles on the study farms. The presence of resistance to these non-utilized classes may be due to co-resistance (linkage of resistance genes) or attributed to AMU earlier in the production chain, such as at the hatchery or broiler breeding farms. This attribution is supported by research in Canada and Belgium, which indicated that increased resistance to ceftiofur could be associated with the use of this antimicrobial within hatcheries [23,24]. This underscores the importance of considering the entire production chain when investigating risk factors for the development and dissemination of AMR in the broiler industry.

At the non-clustered level, we did not find correlations between the proportions of NWT phenotypes to a specific tested antimicrobial when the average use of antimicrobials belonging to the same class increased with one treatment day per cycle at the farm level. A possible explanation could be that the resistance levels found in our study (both proportions of NWT phenotypes and proportions of clinical resistance) are already relatively high for the most used antimicrobial classes, implying that relatively small changes in AMU would not result in significant changes in susceptibility patterns. This is similar to findings from a comparable study conducted on broiler farms in Bangladesh, where the correlation between AMU and AMR was weak or absent [25]. The possible explanation provided was that in a population with relatively low levels of AMR, small increases in AMU could result in significantly larger shifts in resistance compared to similar changes in populations where resistance levels are already high [25]. Moreover, AMU and AMR surveillance data from the Netherlands indicate a lag between reductions in AMU and observable decreases in AMR [26]. This highlights an important consideration when aiming to reduce AMR: reductions in AMU may not lead to immediate decreases in resistance levels, particularly if resistance is already high.

Comparing the AMU levels we found on Indonesian broiler farms in our study with the results derived from other recent studies conducted in the poultry sector is challenging. Previous studies on AMU and AMR in the poultry sector in Southeast Asia often relied on cross-sectional AMU questionnaire data, which cannot always be validated [27]. Additionally, the metric used to express AMU differs considerably between studies, making direct comparisons difficult. Studies in Vietnam and Pakistan expressed AMU in mg/kg live body weight [28,29], while a study in the Philippines used the number of active antimicrobial ingredients being used (AAIs) [30]. One study in Bangladesh expressed AMU using the dose-based indicator veterinary defined daily dose (DDDvet) as defined by the European Surveillance of Veterinary Antimicrobial Consumption (ESVAC) [31,32]. However, as mentioned by the authors, this dosage is not the standardized dose in Bangladesh, possibly under- or overestimating the treatment frequency [32].

Furthermore, production systems differ between countries; for example, differences in production cycle length can significantly affect AMU calculations, complicating quantitative comparisons of AMU between countries [33,34]. This variability in measurement methods underscores the challenge of comparing AMU across different studies and regions. Nevertheless, a qualitatively consistent finding across the studies mentioned above is the relatively high use of (HP)CIAs, such as amoxicillin, enrofloxacin, colistin, and erythromycin, which we also found in our study.

On the 19 farms included in this study, AMU exhibited considerable variability in both the quantities and types of antimicrobials that were used (Figure 2; Table 2). While penicillins and quinolones were commonly utilized across most farms, aminoglycosides were used by only a few, and phosphonic acid derivates were used on only one farm (farm 9 (Appendix A)). Although some farms reported having a standard treatment protocol, this did not consistently result in a standardized treatment regimen. This inconsistency suggests that protocols may have varied across cycles or were not strictly followed. Additionally, deviations from the protocol may have occurred due to disease outbreaks on the farm, prompting farmers to alter their usual treatment practices. Regardless of the origin, the variability indicates that using relatively low amounts of antimicrobials on small- to medium-scale broiler farms in Indonesia is feasible. Farmers with high AMU could potentially benefit from adopting practices used by those with lower AMU.

In our analysis, several VMPs used on the included farms contained a combination of an HPCIA with an antimicrobial of lesser importance to human health, either a CIA, HIA, or IA (Figure 1). Given that a high percentage of AMU in the Indonesian broiler sector is reported as preventive (80%), there is an opportunity for all parties involved to reconsider whether combination products are truly necessary [14,15]. If antimicrobials must be used preventively, a single product containing an antimicrobial of lesser importance may be more appropriate.

This concern is further emphasized by the high preventive use of HPCIAs on our study farms. Macrolides, quinolones, and polymyxins were commonly used across the majority of farms, mostly, if not exclusively, for preventive purposes. This highlights the need to critically assess the current antimicrobial treatment practices and protocols. In alignment with the objectives of the NAP to promote prudent AMU in livestock, the Indonesian government banned the use of colistin in livestock in December 2019 [35]. Although colistin was still frequently used in this dataset, which was partly collected before the ban of colistin, it would be valuable to measure AMU in current broiler production cycles to assess the impact of this regulation.

There were certain limitations in our AMU data collection. One limitation to consider is the relatively small sample size. While data were collected across 78 production cycles, these cycles originated from only 19 farms situated in Wes Java. Consequently, our findings are not fully representative of the entire broiler sector in Indonesia. Nonetheless, the advantages of a longitudinal study design should not be overlooked. In many studies on AMU or AMR in poultry, data are collected cross-sectionally, relying on a single data collection point. By intensively monitoring a smaller group of farms over an extended period, we can more accurately identify specific patterns in AMU and management practices, enabling the development of targeted interventions. This approach, however, limits data collection to a smaller scale due to its intensive nature. We strongly recommend that future research apply this data collection method on a larger scale to obtain more representative data for the region.

Ideally, the downtime between production cycles (the period when the houses are vacant between production cycles) would be similar across different farms for the purpose of analysis, ensuring a comparable time effect for variations in AMU and its potential correlation with AMR. However, in practice, it was not possible to achieve such uniformity. For many farmers, it was common practice to extend or shorten the downtime period to allow for some variations in farm management practices and respond to fluctuations in the price of day-old chicks (DOCs). The potential differences in the impact of AMU on AMR during the early production cycles compared to later cycles closer to the time of sampling remain uncertain. No evidence could be found to adequately account for the time effect in poultry. Moreover, a risk of shortening the downtime is that the cleaning and disinfection routine cannot be performed as rigorously. Studies from Thailand and Norway warn that inadequate biosecurity measures during downtime can result in the persistence of resistant *E. coli* strains [36,37].

Another limitation was that the farmers in our study were not accustomed to maintaining treatment records or disposing of all packaging in the designated waste bins. Although the extension workers visited the farms three times per production cycle and maintained weekly contact with the farmers, we cannot fully guarantee that some packaging was not misplaced or that treatment records were not completed differently from actual practices. Finally, the data were manually transferred from the treatment records to a consolidated dataset for analysis. As noted in the guidelines for measuring AMU, manual data transfer carries the risk of input errors [34]. This could be minimized in future studies by ensuring that farmers record their AMU in a format directly usable for analysis, such as an electronic form that can be easily transferred into software for analysis. In our study, using a digital form was not feasible due to the remote location of the farms, where limited internet access and the absence of computers were common. Developing an easy-to-use mobile phone interface could potentially facilitate the use of electronic forms in such settings.

AMR levels across the 25 collected isolates per study farm varied greatly per antimicrobial. When ECOFFs were used, NWT phenotypes were found for 6 out of 14 tested antimicrobials, with the highest proportions of NWT being for ciprofloxacin (93%, HPCIA), ampicillin (88%, CIA), and tetracycline (83%, HIA) (Table 3). Monitoring these levels of NWT phenotypes is important for tracking changes in the susceptibility to antimicrobials. These changes should be monitored as a decrease in susceptibility can become a public health threat for multiple reasons. Resistant bacteria and/or resistance genes can be transferred from broilers to farmers through direct or indirect contact and can be transmitted to consumers through contamination of poultry meat in slaughterhouses [38,39]. A less visible but equally important public health threat can arise through the consumption of vegetables grown on land that was fertilized using broiler feces, a relatively common practice [38,39]. In Indonesia, many small-scale vegetable farmers use chicken manure as fertilizer due to its low transportation costs and easy accessibility and handleability [40]. Clearly, much is to be gained from increased awareness and measures to reduce AMU and AMR in Indonesia considering the reliance of the human population on the large poultry sector for their livelihood and food security. Moreover, prudent AMU and reduced AMR have an impact on food security and public health that extends beyond country borders.

As a final note, verifying which breakpoints were employed to analyze antimicrobial resistance trends is crucial. Differences in AMR interpretation can arise depending on whether CBP or ECOFF values are used, and whether these are derived from the Clinical and Laboratory Standards Institute (CLSI) or the European Committee on Antimicrobial Susceptibility Testing (EUCAST). Such variations can lead to differences in observed resistance levels. Although the use of CBPs is often justified by their ease of interpretation for veterinarians, it is important to recognize that the objective of surveillance studies may not solely be to support clinical interpretation; it is also aimed at monitoring changes in susceptibility patterns from a one-health perspective [41]. This is particularly relevant when using indicator bacteria, such as commensal *E. coli*, where the representativeness of CBPs measured in these isolates for treatment efficacy in pathogenic *E. coli* strains may be uncertain [41]. In our study, we deliberately chose to use ECOFF values as our objective was to assess the prevalence of NWT bacteria at the farm level. While each method for evaluating AMR has its merits, it is essential to select the appropriate cut-off values based on the study’s objectives. Comparing results from studies that use different breakpoints—whether CBP or ECOFF, and whether derived from CLSI or EUCAST—can lead to misleading conclusions. Therefore, the sampling methods and breakpoints used must be clearly specified, and raw MIC data must be provided. This ensures accurate comparisons between studies that may use different resistance interpretation methods.

This study offers insights for policymakers aiming to address AMR on small- and medium-scale broiler farms in Indonesia. Although our sample size is limited, the use of HPCIAs, such as enrofloxacin, points to the need for further exploration of AMU across more diverse farms. The variation in AMU between production cycles suggests that longitudinal data collection might provide a clearer understanding of farm-level practices compared to cross-sectional surveys, aiding efforts to monitor and reduce AMU over time. Part of our data were collected before the ban on colistin, and evaluating the effects of this policy change could yield useful insights. Engaging all stakeholders in the broiler production chain, including hatcheries and breeder farms, may also contribute to more effective AMU monitoring and reduction.

## 4. Materials and Methods

This study is part of a larger research project (CORNERSTONE) that ran between 2019 and 2023 [42]. An overview of the study design can be found in Figure 6. The study was conducted on 19 broiler farms in West Java, Indonesia, focusing on the relationship between AMU and AMR in commensal *E. coli* isolates. Data were collected across multiple production cycles to assess AMU patterns and the susceptibility of *E. coli* isolates to fourteen antimicrobials. Statistical analyses, including Spearman correlation and a generalized linear mixed model, were used to evaluate the correlation between AMU and AMR at both farm and antimicrobial class levels.

### 4.1. Farm and Sample Selection

For this study, we selected small- and medium-scale farms in West Java, Indonesia. This is because this group comprises the largest number of commercial farms in Indonesia [15]. The study focused on West Java because most of Indonesia’s broiler farms are located in this region [15]. Farms were selected using a convenience sampling method from the client database provided by the Indonesian veterinary pharmaceutical company Medion. The selection criteria were that the farms were either contract or independent farms, located in West Java, and had a small (housing >5000 and ≤50,000 broilers) to medium scale (housing >50,000 and ≤1,000,000 broilers) [43]. According to the Food and Agriculture Organization of the United Nations (FAO), most of these farms belong to Sector 2, which is defined as commercial-scale broiler production systems where broilers are kept indoors, with flock sizes typically exceeding 10,000 [44,45].

During the recruitment process, farmers were informed that the objective of the project was to gain insights into on-farm AMU and AMR in order to develop recommendations to optimize AMU. All farmers signed an informed consent form before data collection and could withdraw from the study at any time. All data collected on each broiler farm were anonymized.

For this study, commensal *E. coli* was used as an indicator organism for susceptibility testing. For each farm, 25 *E. coli* isolates were tested for antimicrobial susceptibility. This number was based on earlier sample size calculations performed by Persoons et al. (2011), who employed bootstrapping techniques to determine the optimal sample size for the accurate estimation of AMR levels in an epidemiological unit [46]. This method also takes into account the different prevalences of resistance for the different antimicrobials tested.

### 4.2. Antimicrobial Usage Data Collection

AMU data were collected on every farm from a minimum of four successive production cycles of one representative broiler house. Where the farm schedule allowed, AMU data were collected from five production cycles. The production cycles, i.e., the rearing period of broilers from DOC to slaughter, had an average duration of 30 days. At the beginning of the study, farmers were instructed by extension workers from the Center for Indonesian Veterinary Analytical Studies (CIVAS) on the specific data they needed to collect. These data included daily treatment records and used VMP package collection in specially assigned drug bins. Treatment records included the date and age of the broilers at time of application, the (brand) name of the VMP, purpose of use, the amount of product used (either in (milli)liters, (milli)grams, or number of packets), and the application route. To optimize data collection quality, the extension workers visited each participating farm three times per production cycle and stayed in close contact with the farmers about the situation on the farm and the progress of data collection.

AMU can be quantified using different indicators. In the context of the farms included in this study, a count-based indicator seemed most suitable, as described in a previous publication [33]. A count-based indicator counts the treatment days in the numerator and divides this by the period at risk. In this study, we used the count-based treatment frequency (TF_count-based_), which was calculated as follows:(1)$$\frac{number\;of\;treatment\;days\;of\;active\;substance\;per\;cycle}{average\;length\;of\;a\;production\;cycle}$$

This resulted in a proportion of treatment days. This means that if a farm had a TF_count-based_ of 0.5, the broilers at this farm had 15 (30 × 0.5) days of antimicrobial treatment in one standardized production cycle of 30 days. Antimicrobial products containing two active substances were counted as two separate treatments.

### 4.3. Data Collection of Commensal Escherichia coli (E. coli) Isolates for Susceptibility Testing

Sampling was carried out 1–2 days before the slaughter of the final production cycle from which the usage data were collected. The boot swab method was used for sample collection in the following way. An extension worker from CIVAS walked at least 100 steps in a zigzag pattern through the broiler study house while wearing boots covered by sterilized boot covers (https://antonides.com/products/overschoenen-pp-steriel?_pos=1&_psq=sterile+over&_ss=e&_v=1.0). Every 1/5th of the house, the litter sample attached to the boot cover was collected in a plastic bag using a wooden tongue depressor. This resulted in one sample per farm, with a required minimum weight of 25 g of feces combined with litter.

Within three hours, the samples were transported in a cool box to the laboratory (National Quality Control Laboratory for Animal Product Testing and Certification—BPMSPH, Bogor, Indonesia). Upon arrival, the samples were either processed on the same day or stored overnight at 4 °C for processing the next day. The samples were directly plated onto five separate MacConkey agar plates with the aim of obtaining at least 25 distinct *E. coli* colonies. The plates were incubated for 18–24 h at 37 °C. Phenotypically suspected *E. coli* colonies were subcultured and confirmed as *E. coli* after a positive indole test. The protocol applied for susceptibility testing is in line with the WHO-approved Tricycle protocol [47]. These isolates were then subcultured on MacConkey agar plates (Oxoid, Basingstoke, Hampshire, United Kingdom) and stored in 20–30% glycerol at −80 °C.

The susceptibility profiles of 25 *E. coli* isolates per farm were assessed using Sensititre Minimum Inhibitory Concentration EUVSEC plates (Thermo Fisher Scientific, Landsmeer, the Netherlands) [48]. This was carried out for a panel of 14 antimicrobials: sulfamethoxazole, trimethoprim, ciprofloxacin, tetracycline, meropenem, azithromycin, nalidixic acid, cefotaxime, chloramphenicol, tigecycline, ceftazidime, colistin, ampicillin, and gentamicin (Appendix A).

Since commensal *E. coli* was used as the indicator bacteria, the prevalence of antimicrobial NWT was assessed using ECOFF values established by EUCAST [21]. Although CBPs are typically determined for specific (pathogenic) bacteria in a particular host, many studies in Southeast Asia on poultry use human CBPs to assess resistance levels in commensal (non-pathogenic) *E. coli* regardless of clinical efficacy or host specificity [49,50]. We included the human CBPs in our analysis to allow for easier comparison with these regional studies. The CBPs were derived from the CLSI [22]. Both ECOFF and CBP data were obtained from their respective sources in September 2024. For tigecycline, the CBP from EUCAST was used because no CBP was available from CLSI.

### 4.4. Data Analysis and Statistics

All collected data were analyzed using R version 2023.9.1.494 [51]. The packages used were tidyr, tidyverse, dplyr, ggplot2, survival, DHARMa, lme4, glmmTMB, magrittr, cowplot, tibble, and reshape2. Descriptive statistics were performed to assess the distribution of the data, followed by analyses of the relationship between AMU and AMR per farm at the clustered and non-clustered levels.

An analysis of the relationship between AMU and AMR per farm was performed at the clustered and non-clustered levels. A clustered analysis using the Spearman correlation coefficient was performed to examine the relationship between the average AMU (averaged for all antimicrobial classes per cycle per farm, expressed as TF_count-based_) and the average number of tested antimicrobials (n = 14) to which *E. coli* isolates from the same farm showed an NWT phenotype (i.e., MIC > ECOFF).

Additionally, a non-clustered analysis was performed for the five antimicrobial classes that were most frequently used on the study farms (fluoroquinolones, macrolides, tetracyclines, polymyxins, and penicillins). The remaining antimicrobial classes were not included because of highly skewed data due to their minimal use on the farms, which precluded drawing meaningful conclusions. The average use of the five most used antimicrobial classes per cycle per farm (expressed as TF_count-based_) was analyzed in relation to the proportion of NWT phenotypes for a tested antimicrobial that belonged to the same antimicrobial class using ECOFFs on the same farm. We employed a generalized linear mixed model (GLMM) with a binomial logistic probability function to evaluate the association between AMU and the occurrence of NWT *E. coli* phenotypes. The dependent variable was the occurrence of NWT *E. coli* phenotypes, the independent variable was AMU of the specific antimicrobial, and we corrected for farm by adding it as a random effect.

## 5. Conclusions

In this study on small- and medium-scale broiler farms in West Java, Indonesia, substantial variability was observed in both the quantity and quality of AMU per farm. Notably, the high percentage of preventive use of HPCIAs highlights the need for farm-level AMU data collection to guide targeted interventions promoting the prudent use of antimicrobials in the future. In line with national surveillance data, AMR was widespread and showed variations between farms. While there was a correlation between overall AMU and the average number of isolates showing NWT phenotypes to the 14 tested antimicrobials, no correlation was found between the use of a specific antimicrobial and resistance. This was probably due to a high baseline prevalence of NWT *E. coli* isolates and relatively few independent observations.

We strongly recommend that further research not only includes more farms but also other parts of the production chain, such as hatcheries and broiler breeding farms. Additionally, this study could be repeated in Indonesia to assess the impact and effectiveness of the ban on colistin use in broiler production, which was introduced in December 2019.

## Figures and Tables

**Figure 1 antibiotics-13-01181-f001:**
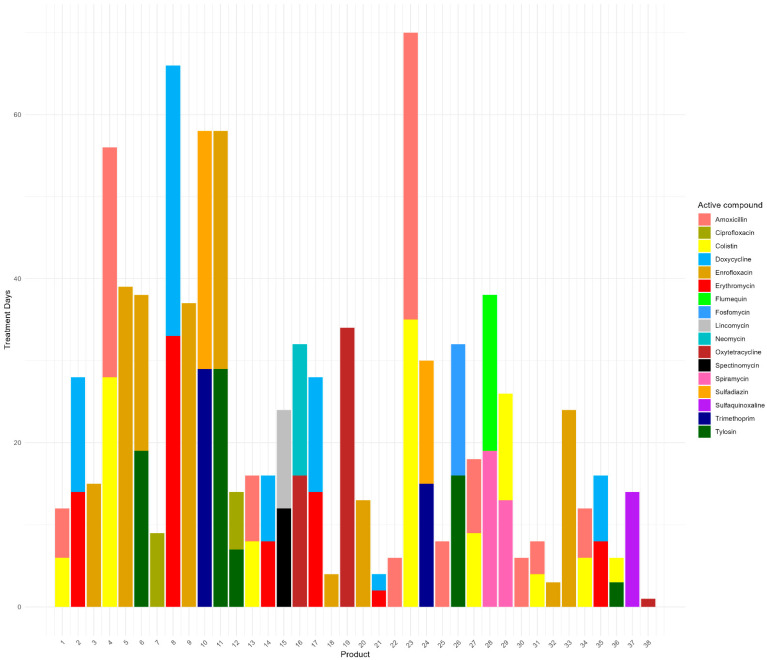
The total number of treatment days per antimicrobial product that was used in the study farms (across 78 production cycles). The product names have been recoded into numbers and are labeled alphabetically.

**Figure 2 antibiotics-13-01181-f002:**
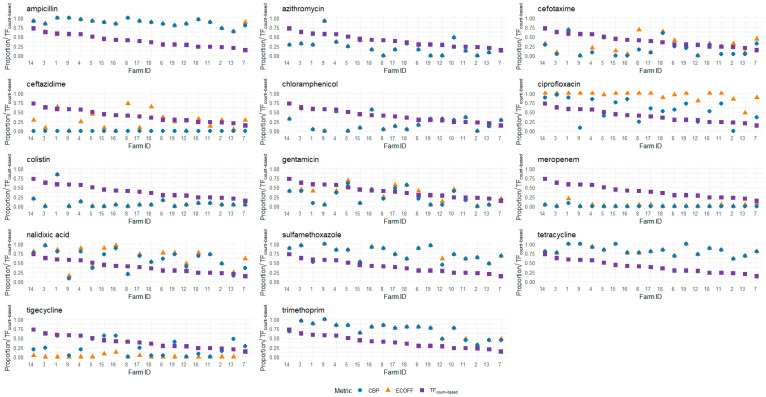
An overview of the average proportions of NWT (ECOFF (orange triangles)) and non-susceptible (CBP (blue circles)) isolates per tested antimicrobial per participating farm, alongside the average AMU per production cycle, expressed in TF_count-based_ (purple squares) on the *y*-axis. The farms are arranged in decreasing order of average AMU per cycle (TF_count-based_). The average AMU was calculated across four or five (only farms 4 and 15) successive production cycles within the study period.

**Figure 3 antibiotics-13-01181-f003:**
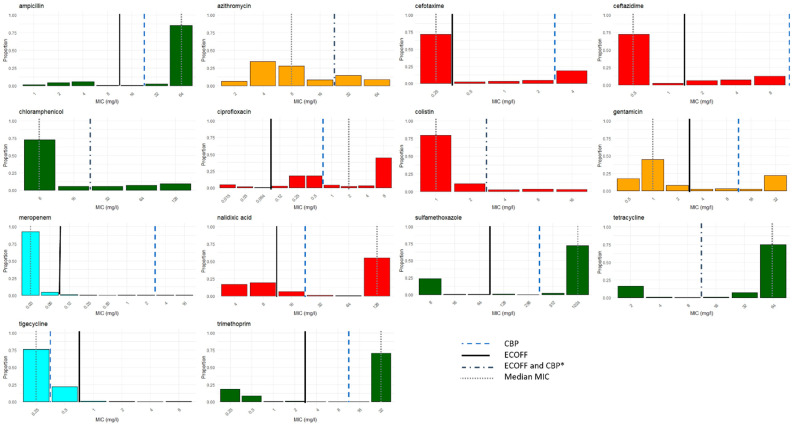
The MIC distribution of isolates across 14 different antimicrobials. The bars are color-coded according to the WHO classifications of antimicrobial importance: the red bars represent HPCIAs, the orange bars denote CIAs, and the green bars signify HIAs. The turquoise bars denote antimicrobials that are categorized as being for human use only. * For azithromycin, chloramphenicol, colistin, and tetracycline, the ECOFF and CBP lines align because the ECOFF breakpoint defines values greater than (>) as indicating NWT, while the CBP breakpoint uses values greater than or equal to (≥) to indicate resistance.

**Figure 4 antibiotics-13-01181-f004:**
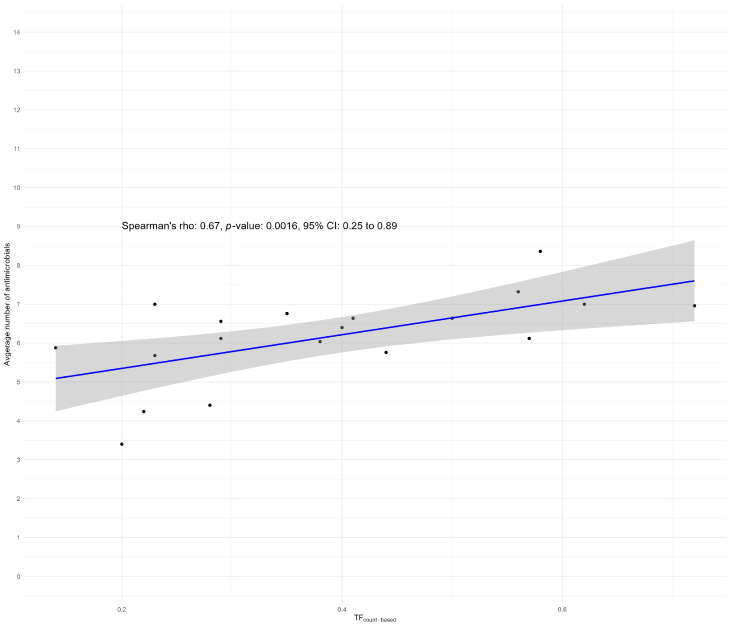
A scatterplot of the Spearman correlation between the average number of antimicrobials for which NWT *E. coli* phenotypes were found in relation to the average AMU (TF_count-based_) over the monitored production cycles per farm.

**Figure 5 antibiotics-13-01181-f005:**
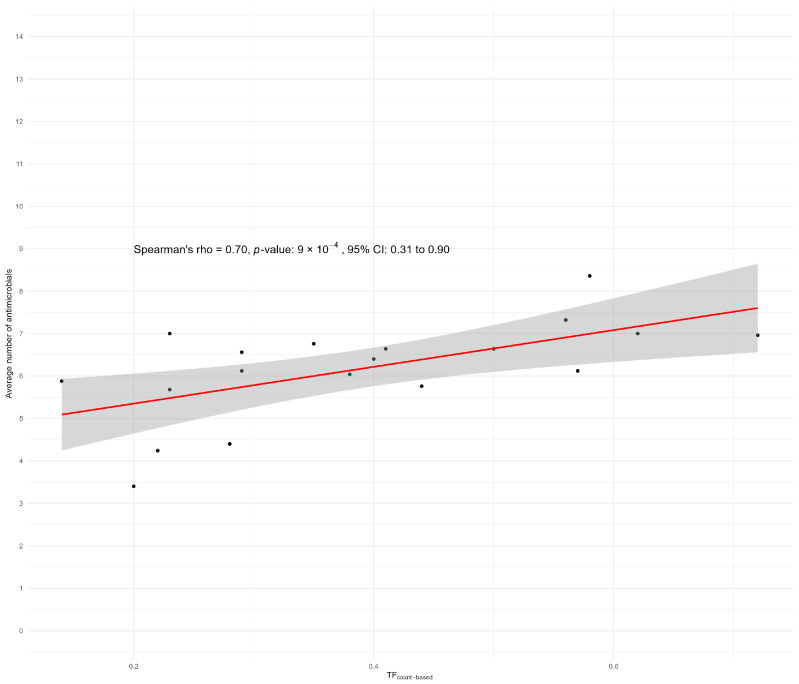
A scatterplot of the Spearman correlation between the AMU in average treatment days per production cycle in relation to the average number of antimicrobials to which isolates tested as resistant according to the CBPs. Because no CLSI CBP for tigecycline was available, the EUCAST CBP was used for tigecycline.

**Figure 6 antibiotics-13-01181-f006:**
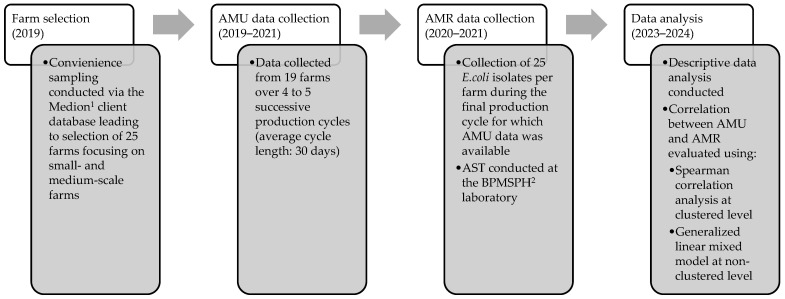
A flowchart illustrating the study workflow, detailing stages from farm selection through AMU and AMR data collection to final data analysis, including correlation assessments between AMU and AMR across clustered and non-clustered levels. ^1^ Medion is an Indonesian veterinary pharmaceutical company. ^2^ BPMSPH is the National Quality Control Laboratory for Animal Product Testing and Certification.

**Table 1 antibiotics-13-01181-t001:** Overview of farm characteristics.

Farm ID	Maximum Capacity of Broilers on Farm	Average Number of Broilers per Study House	Housing System	Use of Standard Treatment Protocol ^1^	Source Standard Treatment Protocol
1	11,000	6000	Open	Yes	Integration
2	12,000	5125	Open	Yes	Integration
3	16,000	7250	Closed	No	No protocol
4	13,500	6750	Semi-closed	No	No protocol
5	50,000	13,900	Closed	Yes	Pharmaceutical company
6	80,000	20,000	Semi-closed	No	No protocol
7	14,000	4625	Open	No	No protocol
8	21,000	4000	Open	No	No protocol
9	70,000	5100	Semi-closed	No	No protocol
10	40,000	1766	Open	Yes	Owner
11	33,000	18,631	Semi-closed	Yes	Owner
12	28,000	7750	Semi-closed	No	No protocol
13	65,000	18,900	Semi-closed	No	No protocol
14	50,000	24,250	Semi-closed	No	No protocol
15	14,000	4369	Open	Yes	Pharmaceutical company
16	6500	4400	Open	No	No protocol
17	4800	3825	Open	No	No protocol
18	40,000	7925	Semi-closed	No	No protocol
19	35,000	13,050	Semi-closed	No	No protocol

^1^ A standard treatment protocol is a predefined plan outlining which treatments should be administered on specific days of the production cycle. In this study, the standard treatment protocol was either provided by an integration (referring to a larger company that the farm is a part of, which oversees and coordinates various aspects of the farm’s operations) or a pharmaceutical company or developed by the farm owner.

**Table 2 antibiotics-13-01181-t002:** Overview of AMU ^1^ per antimicrobial.

Antimicrobial ^1^	Farms with Use (n = 19)	Production Cycles with Use (n = 78)	Total Treatment Days ^2^
Amoxicillin (penicillin (HIA))	15	38	116
Colistin (polymyxin (HPCIA))	14	35	112
Ciprofloxacin (fluoroquinolone (HPCIA))	4	6	16
Doxycycline (tetracycline (HIA))	8	24	79
Enrofloxacin (fluoroquinolone (HPCIA))	12	35	183
Erythromycin (macrolide (CIA))	8	24	79
Flumequine (quinolone (HPCIA))	2	5	19
Fosfomycin (phosphonic acid derivates (HPCIA))	1	3	16
Lincomycin (lincosamide (HIA))	2	3	12
Neomycin (aminoglycoside (CIA))	2	4	16
Oxytetracycline (tetracycline (HIA))	5	13	51
Spectinomycin (aminocyclitol (IA))	2	3	12
Spiramycin (macrolide (CIA))	4	9	32
Sulfadiazine (sulfonamide (HIA))	5	12	44
Sulfaquinoxaline (sulfonamide (HIA))	2	4	14
Trimethoprim (diaminopyrimidine (HIA))	5	12	44
Tylosin (macrolide (CIA))	7	20	74

^1^ For each antimicrobial used, the antimicrobial class and the classification according to the WHO in relation to its importance for human health is added between brackets [16]. In order of decreasing importance to human health, antimicrobials can be classified as highest priority critically important antimicrobials (HPCIAs, red), critically important antimicrobials (CIAs, orange), highly important antimicrobials (HIAs, yellow), or important antimicrobials (IAs, green). ^2^ The sum of treatment days of this specific antimicrobial across 78 production cycles.

**Table 3 antibiotics-13-01181-t003:** Breakpoints and antimicrobial susceptibility of 14 tested antimicrobials.

Tested Antimicrobial ^1^	WHO Classification ^2^	ECOFF ^3^ (mg/L)	CBP ^4^ (mg/L)	% Isolates MIC > ECOFF ^5^	% Isolates MIC ≥ CBP ^6^
Ampicillin (AMP)	Penicillins (aminopenicillins) (HIA)	8	32	88%	88%
Azithromycin (AZI)	Macrolides (CIA)	16	32	23%	23%
Cefotaxime (CEF)	Cephalosporins (3rd and 4th generation) (HPCIA)	0.25	4	28%	19%
Ceftazidime (CEZ)	Cephalosporins (3rd and 4th generation) (HPCIA)	1	16	25%	0%
Chloramphenicol (CHL)	Amphenicols (HIA)	16	32	22%	22%
Ciprofloxacin (CIP)	Quinolones (HPCIA)	0.06	1	93%	54%
Colistin (COL)	Polymyxins (HPCIA)	2	4	9%	9%
Gentamicin (GEN)	Aminoglycosides (CIA)	2	16	30%	25%
Meropenem (MER)	Carbapenems (Human use only)	0.06	4	3%	1%
Nalidixic acid (NAL)	Quinolones (HPCIA)	8	32	64%	57%
Sulfamethoxazole (SUL)	Sulfonamides (HIA)	64	512	75%	74%
Tetracycline (TET)	Tetracyclines (HIA)	8	16	83%	83%
Tigecycline (TIG)	Glycylcycline (Human use only)	0.5	0.5 ^3^	2%	24%
Trimethoprim (TRI)	Diaminopyrimidine (HIA)	2	16	71%	71%

^1^ In order of decreasing importance to human health, antimicrobials can be classified as human use only (turquoise), highest priority critically important antimicrobials (HPCIAs, red), critically important antimicrobials (CIAs, orange), or highly important antimicrobials (HIAs, yellow). ^2^ The WHO classification of medically important antimicrobials according to its importance for human health [16]. ^3^ The ECOFF according to the European Committee on Antimicrobial Susceptibility Testing (EUCAST) [21]. ^4^ The CBP according to the Clinical and Laboratory Standards Institute (CLSI) [22]. Because no CLSI CBP is available for tigecycline, the EUCAST CBP was used. ^5^ The percentage of tested isolates (n = 475) for which the minimal inhibitory concentration (MIC) was higher than the ECOFF value. ^6^ The percentage of tested isolates (n = 475) for which the MIC was higher than or equal to the CBP value.

## Data Availability

The raw data supporting the conclusions of this article will be made available by the authors upon request.

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
