# Peer review of "Antimicrobial Usage and Antimicrobial Resistance in Commensal Escherichia coli from Broiler Farms: A Farm-Level Analysis in West Java, Indonesia"

_antibiotics, 2024, doi:10.3390/antibiotics13121181_

Round 1
Reviewer 1 Report
Comments and Suggestions for Authors
1. I would like to express my gratitude for the opportunity to review the manuscript entitled “Antimicrobial usage and antimicrobial resistance in commensal Escherichia coli from broiler farms: A farm-level analysis in West Java, Indonesia.” The study is well-written and presents a thoughtfully designed set of experiments that contribute valuable insights to the field. However, there are some concerns that must be addressed in order to be considered for publication.
2. I suggest removing the "List of Abbreviations" section, as it may not be necessary for clarity. In academic writing, it's more effective to write the full term followed by the abbreviation in parentheses, which enhances readability while maintaining a formal style.
3. Since the sampling timeframe and the farm sites were not mentioned, I have concerns about the limited spatio-temporal scope of your sample collection. Collecting samples from only 25 small- and medium-scale broiler farms in West Java, Indonesia, may not provide a comprehensive representation of antimicrobial resistance.
4. While protocol for identifying E. coli using MacConkey agar and an indole test is a good starting point, I believe it has some limitations. MacConkey agar may allow non-pathogenic lactose fermenters to grow, potentially leading to misidentification, and relying solely on the indole test may not provide sufficient confirmation. I recommend incorporating additional tests, such as the methyl red, Voges-Proskauer, and citrate utilization tests, as well as assessing colony morphology from a wider selection of colonies to enhance the accuracy of your identification process.
5. I would like to point out that Trimethoprim is classified as a diaminopyrimidines group. Please revise the description to accurately reflect this classification.
6. I suggest that the author thoroughly addresses the study's limitations in the discussion section. Acknowledging and discussing these limitations will enhance the transparency and interpretation of the findings, improving the overall quality of the manuscript.
Author Response
Thank you for reviewing our manuscript and providing the opportunity for submitting a revised manuscript. Please find below our response to the comments and suggestions of the reviewers. All adjustments have been made using track changes in the manuscript. During the language editing process by a professional English editor / native speaker, and thoroughly reviewing the entire paper again we have made some additional minor changes to the text. These were not all requested as such by the reviewers, but were made because we thought that these changes would make the manuscript more clear.
Important note for clarity: The line numbers are cited as they appear when track changes are expanded. This coincides with the line numbers in the attached word document. In the PDF document, the track changes are compressed, and the line numbers are different.
Comment 1: I would like to express my gratitude for the opportunity to review the manuscript entitled “Antimicrobial usage and antimicrobial resistance in commensal Escherichia coli from broiler farms: A farm-level analysis in West Java, Indonesia.” The study is well-written and presents a thoughtfully designed set of experiments that contribute valuable insights to the field. However, there are some concerns that must be addressed in order to be considered for publication.
Response 1: Thank you for your efforts to review our manuscript. We appreciate your thoroughness and feedback to improve the manuscript. Below you will find the adjustments we’ve made based on your suggestions.
Comment 2: I suggest removing the "List of Abbreviations" section, as it may not be necessary for clarity. In academic writing, it's more effective to write the full term followed by the abbreviation in parentheses, which enhances readability while maintaining a formal style.
Response 2: Thank you for your suggestion. We have thoroughly reviewed the manuscript to ensure that each abbreviation is initially introduced by presenting the full term, followed by the abbreviation in parentheses. Given the manuscript’s length, we believe that adding a list of abbreviations will enhance readability by providing readers—particularly those less familiar with certain abbreviations—an accessible reference to quickly understand meanings without needing to search for each initial mention within the text. Also, removing it would go against the preference of one of the other reviewers, hence we decided it would be best to keep the list in the manuscript.
Comment 3: Since the sampling timeframe and the farm sites were not mentioned, I have concerns about the limited spatio-temporal scope of your sample collection. Collecting samples from only 25 small- and medium-scale broiler farms in West Java, Indonesia, may not provide a comprehensive representation of antimicrobial resistance.
Response 3: Response 3: We agree with you that 19 farms, which is the number of farms included in the final analysis, is a relatively limited sample size although we analysed a total of 78 separate production cycles. We have included a paragraph in the discussion section that more clearly mentions the limited sample size to provide transparency in Line 374 – Line 386. We also discussed the time effect of variations in AMU and its potential correlation with AMR in the subsequent paragraph Line 387 – Line 402. The data collection took place between 2019 and 2021. To provide more clarity and as suggested by another reviewer we’ve added a new figure (Figure 6) to visualize the process of data collection. We believe this sufficiently illustrates the workflow of our study and provides transparency.
Comment 4: While protocol for identifying E. coli using MacConkey agar and an indole test is a good starting point, I believe it has some limitations. MacConkey agar may allow non-pathogenic lactose fermenters to grow, potentially leading to misidentification, and relying solely on the indole test may not provide sufficient confirmation. I recommend incorporating additional tests, such as the methyl red, Voges-Proskauer, and citrate utilization tests, as well as assessing colony morphology from a wider selection of colonies to enhance the accuracy of your identification process.
Response 4: We realize that a more extensive biochemical testing could have been useful if the aim was to differentiate lactose fermenting bacterial species. However, in this case we selected suspected E. coli colonies (typical red/pinkish colonies) as it was our aim to use only E. coli for further testing. Based on our longstanding experience, the phenotypic selection in combination with a positive indol assay, very rarely leads to a wrong identification (in a very exceptional case an Enterobacter – 1/100 strains). We follow the recommendations given in the WHO approved Tricycle protocol that describes the indol test as the sole confirmation test for MacConkey suspected E. coli colonies (9789240021402-eng.pdf). We do not specifically include pathogenic E. coli. Most of the E. coli isolated from the droppings of healthy animals are assumed to be non-pathogenic and we therefore refer to them as ‘commensal E. coli’. For transparency, we have also added the reference to the Tricycle protocol in Line 554 (Reference number 49).
Comment 5: I would like to point out that Trimethoprim is classified as a diaminopyrimidines group. Please revise the description to accurately reflect this classification.
Response 5: Our apologies for our oversight. We have adjusted the classification for Trimethoprim (Table 2: Overview of AMU per antimicrobial, and Table 3: Breakpoints and antimicrobial susceptibility for 14 tested antimicrobials)
Comment 6: I suggest that the author thoroughly addresses the study's limitations in the discussion section. Acknowledging and discussing these limitations will enhance the transparency and interpretation of the findings, improving the overall quality of the manuscript.
Response 6: Thank you for the suggestion. In Line 374 – Line 384 we have added a paragraph discussing the limitation of our sample size. We believe this addition enhances the transparency and interpretation of our findings by directly addressing the constraints of our sample. In the subsequent paragraphs Line 385 – Line 417, we further examine other study limitations. Upon careful review, we are confident that this section of the discussion provides a thorough and transparent account of our study’s limitations.
Reviewer 2 Report
Comments and Suggestions for Authors
1. Lines 400-413, should be deleted. This is unnecessary in the method. If required a few sentences should insert later subheading in the method section..
2. The research identified E. coli only through the MacConkey Agar plate without Molecular techniques.. Many other species can be grown in the same plate. Therefore, only molecular genetic identification is essential now a days to declare the strains.
3. grammatical errors should be checked
Comments on the Quality of English Language
English language need to be improved
Author Response
Thank you for reviewing our manuscript and providing the opportunity for submitting a revised manuscript. Please find below our response to the comments and suggestions of the reviewers. All adjustments have been made using track changes in the manuscript. During the language editing process by a professional English editor / native speaker, and thoroughly reviewing the entire paper again we have made some additional minor changes to the text. These were not all requested as such by the reviewers, but were made because we thought that these changes would make the manuscript more clear.
Important note for clarity: The line numbers are cited as they appear when track changes are expanded. This coincides with the line numbers in the attached word document. In the PDF document, the track changes are compressed, and the line numbers are different.
Comment 1: Lines 400-413 (in the revised manuscript this is Line 471 – Line 478), should be deleted. This is unnecessary in the method. If required a few sentences should insert later subheading in the method section.
Response 1: Thank you for taking the time to critically review our manuscript. Based on your suggestion and the suggestion by Reviewer 3 we have restructured the Materials and Methods section to increase clarity and readability. Lines 400-413 (or in the revised manuscript this is Line 471 – Line 478), have been rephrased in Line 471 to Line 478.
Comment 2: The research identified E. coli only through the MacConkey Agar plate without Molecular techniques.. Many other species can be grown in the same plate. Therefore, only molecular genetic identification is essential now a days to declare the strains.
Response 2: We realize that a more extensive biochemical testing could have been useful if the aim was to differentiate lactose fermenting bacterial species. However, in this case, we selected suspected E. coli colonies (typical red/pinkish colonies) as it was our aim to use only E. coli for further testing. Based on our longstanding experience, the phenotypic selection in combination with a positive indole assay very rarely leads to a wrong identification. We follow the recommendations given in the WHO-approved Tricycle protocol that describes the indole test as the sole confirmation test for MacConkey suspected E. coli colonies (9789240021402-eng.pdf). We do not specifically include pathogenic E. coli. Most of the E. coli isolated from the droppings of healthy animals are assumed to be non-pathogenic, and we therefore refer to them as “commensal E. coli.”
In addition, most laboratories in LMIC still use phenotypic methods, as it is too expensive to use molecular identification. The economic constraints in LMIC settings make phenotypic methods the most feasible and accessible option. For transparency, we have added a reference to the Tricycle protocol (reference number 49) in the revised manuscript at Line 554.
Comment 3: Grammatical errors should be checked.
Response 3: Based on your feedback and that of the other reviewers, we have thoroughly checked the manuscript for grammatical errors. In addition, we had the manuscript thoroughly reviewed by a professional English editor. Comprehensive edits were made to ensure grammatical accuracy, and these changes have been applied throughout the text, as reflected in the track changes, to the best of our ability.
Reviewer 3 Report
Comments and Suggestions for Authors#1: Re-check the entire English ... The English is approximately correct, but some problems need to be checked, and complex sentences should be simplified at this time. Let me mention concrete examples.
#1-1 In the introduction, you wrote: Antimicrobial resistance (AMR) forms a major public health threat and was associated with 4.95 million human deaths in 2019, with resistant bacteria developing and spreading among livestock, driven by antimicrobial use (AMU), and creating a potential reservoir for humans through direct contact, the food chain, or the environment. But you can modify the part as follows.
Antimicrobial resistance (AMR) is a serious public health threat, linked to 4.95 million human deaths in 2019. Resistant bacteria can develop and spread in livestock due to antimicrobial use (AMU), creating a potential reservoir that can affect humans through direct contact, the food chain, or the environment.
#1-2: You wrote in the discussion part.
While we observed a significant correlation between the average antimicrobial use (AMU) per farm and the prevalence of non-wildtype (NWT) E. coli isolates for multiple antimicrobials, this correlation was less clear for specific classes due to the high baseline prevalence of NWT isolates, likely influenced by long-standing AMU practices. But you can modify the part as follows.
We observed a significant correlation between average antimicrobial use (AMU) per farm and the prevalence of non-wildtype (NWT) E. coli isolates for multiple antimicrobials. However, this correlation was less clear for specific classes, likely due to the high baseline prevalence of NWT isolates from long-term AMU practices.
#2: Some figures and tables could benefit from clearer labeling or simplification. For instance, using additional legends or more distinct color coding may improve the reader’s understanding of complex data.
-
Figure 2 (Overview of average proportions of NWT and non-susceptible isolates)
- Improvement Suggestion: The figure currently displays different markers (green triangles, coral circles, blue squares) representing various data points. Adding a detailed legend that explicitly explains each marker's meaning (e.g., NWT isolates, non-susceptible isolates, average AMU per cycle) would clarify the relationships. Additionally, using distinct color shades or thicker lines to separate each data type would improve readability.
- Proposed Addition: A legend could be added below the figure explaining each color and shape (e.g., “Green triangles: NWT isolates per antimicrobial; Coral circles: Non-susceptible isolates per antimicrobial; Blue squares: Average AMU per cycle”). This will ensure readers understand the data points at a glance.
-
Table 2 (Overview of AMU per Antimicrobial)
- Improvement Suggestion: This table lists several antimicrobials alongside farms and production cycles, which can be visually overwhelming. Highlighting or color-coding the classes of antimicrobials (e.g., HPCIAs, CIAs) would help differentiate the data. Adding a header row with explanations for each category (e.g., Total Treatment Days, Farms with Use) would make the table easier to interpret.
- Proposed Addition: Include a row at the top with definitions for abbreviations and terms, such as “HPCIA” or “CIA.” Additionally, use a soft background color to distinguish each antimicrobial class, helping readers quickly identify patterns in antimicrobial use.
- #3: Briefly define or explain technical terms, especially those related to antimicrobial classifications or specific statistical measures, for accessibility to a broader audience. I recommend you to make a table where some specific terms should be explained. Concrete speaking, I can mention the technical terms as follows.
Briefly define or explain technical terms, especially those related to antimicrobial classifications or specific statistical measures, for accessibility to a broader audience. - #4: Reorganize the methodology details, so that the readers could understand effectively. To achieve the goal, you should modify your original text according to the following tips.
- Add a Brief Overview Paragraph:
- At the beginning of the Methods section, include a short overview paragraph summarizing the study’s main stages. This could provide a high-level view of the study design, sample collection, AMU and AMR measurement, and data analysis. For instance:
- Example Overview:
- “This study was conducted on 19 broiler farms in West Java, Indonesia, focusing on the relationship between antimicrobial use (AMU) and antimicrobial resistance (AMR) in E. coli isolates. Data was collected across multiple production cycles to assess AMU patterns and susceptibility of E. coli isolates to various antimicrobials. Statistical analyses, including Spearman correlation and generalized linear mixed models, were used to evaluate the correlation between AMU and AMR at both farm and antimicrobial class levels.”
- Insert Subheadings for Key Study Steps:
- Breaking down the section with clear subheadings would guide readers through the process in an organized way. Suggested subheadings might include:
- “Farm and Sample Selection”
- “Data Collection on Antimicrobial Use (AMU)”
- “Isolation and Susceptibility Testing of E. coli”
- “Data Analysis and Statistical Methods”
- Add a Simplified Flowchart:
- Include a flowchart at the beginning of the section to visually represent the study’s workflow. This could show the progression from farm selection to data analysis in a straightforward, visual format, allowing readers to grasp the methodology quickly before reading the detailed descriptions.
- #5 Please include a short section in the discussion about the potential policy and global health implications of this study would enhance the manuscript’s impact, especially for readers from different fields.
I feel the English is generally good, But minor revisions for sentence structure, readability, and occasional term explanations could enhance the document's clarity and accessibility.
Author Response
Thank you for reviewing our manuscript and providing the opportunity for submitting a revised manuscript. Please find below our response to the comments and suggestions of the reviewers. All adjustments have been made using track changes in the manuscript. During the language editing process by a professional English editor / native speaker, and thoroughly reviewing the entire paper again we have made some additional minor changes to the text. These were not all requested as such by the reviewers, but were made because we thought that these changes would make the manuscript more clear.
Important note for clarity: The line numbers are cited as they appear when track changes are expanded. This coincides with the line numbers in the revised manuscript attached as a word document. In the PDF document, the track changes are compressed, and the line numbers are different.
Comment 1: Re-check the entire English. The English is approximately correct, but some problems need to be checked, and complex sentences should be simplified at this time. Let me mention concrete examples.
- 1-1 In the introduction, you wrote: Antimicrobial resistance (AMR) forms a major public health threat and was associated with 4.95 million human deaths in 2019, with resistant bacteria developing and spreading among livestock, driven by antimicrobial use (AMU), and creating a potential reservoir for humans through direct contact, the food chain, or the environment.
But you can modify the part as follows:
Antimicrobial resistance (AMR) is a serious public health threat, linked to 4.95 million human deaths in 2019. Resistant bacteria can develop and spread in livestock due to antimicrobial use (AMU), creating a potential reservoir that can affect humans through direct contact, the food chain, or the environment. - 1-2: You wrote in the discussion part.
While we observed a significant correlation between the average antimicrobial use (AMU) per farm and the prevalence of non-wildtype (NWT) coli isolates for multiple antimicrobials, this correlation was less clear for specific classes due to the high baseline prevalence of NWT isolates, likely influenced by long-standing AMU practices.
But you can modify the part as follows:
We observed a significant correlation between average antimicrobial use (AMU) per farm and the prevalence of non-wildtype (NWT) E. coli isolates for multiple antimicrobials. However, this correlation was less clear for specific classes, likely due to the high baseline prevalence of NWT isolates from long-term AMU practices.
Response 1: Thank you for your feedback and for adding the concrete suggestions. We have revised the sections from your first example according to your suggestion in Line 75 – Line 80. The sentence of the second example, has been rewritten, as well. In the rewritten form, it still conveys our main message, but was adapted by a professional English editor. The entire manuscript has been thoroughly reviewed and edited by this native speaker. Comprehensive edits were made to ensure grammatical accuracy, and these changes have been applied throughout the text, as reflected in the track changes, to the best of our ability.
Comment 2: Some figures and tables could benefit from clearer labelling or simplification. For instance, using additional legends or more distinct colour coding may improve the reader’s understanding of complex data.
- Figure 2 (Overview of average proportions of NWT and non-susceptible isolates)
- Improvement Suggestion: The figure currently displays different markers (green triangles, coral circles, blue squares) representing various data points. Adding a detailed legend that explicitly explains each marker's meaning (e.g., NWT isolates, non-susceptible isolates, average AMU per cycle) would clarify the relationships. Additionally, using distinct color shades or thicker lines to separate each data type would improve readability.
- Proposed Addition: A legend could be added below the figure explaining each color and shape (e.g., “Green triangles: NWT isolates per antimicrobial; Coral circles: Non-susceptible isolates per antimicrobial; Blue squares: Average AMU per cycle”). This will ensure readers understand the data points at a glance.
- Table 2 (Overview of AMU per Antimicrobial)
- Improvement Suggestion: This table lists several antimicrobials alongside farms and production cycles, which can be visually overwhelming. Highlighting or color-coding the classes of antimicrobials (e.g., HPCIAs, CIAs) would help differentiate the data. Adding a header row with explanations for each category (e.g., Total Treatment Days, Farms with Use) would make the table easier to interpret.
- Proposed Addition: Include a row at the top with definitions for abbreviations and terms, such as “HPCIA” or “CIA.” Additionally, use a soft background color to distinguish each antimicrobial class, helping readers quickly identify patterns in antimicrobial use.
Response 2: Thank you for the concrete suggestions to improve readability and interpretation of Figure 2 and Table 2.
For Table 2, we have applied colour coding to provide more clarity. In addition we have elaborated on the different classes in the first footnote of the table in Line 180 – Line 183.
For Figure 2, we implemented your suggestion by using more distinct colors (blue, orange, and purple). Additionally, we simplified the figure by placing the legend for the metrics at the bottom, rather than repeating it multiple times.
To further improve readability of the figures. We have also adjusted the y-axis description of Figure 4 and Figure 5.
Comment 3: Briefly define or explain technical terms, especially those related to antimicrobial classifications or specific statistical measures, for accessibility to a broader audience. I recommend you to make a table where some specific terms should be explained. Concrete speaking, I can mention the technical terms as follows.
Response 3: We have reviewed the manuscript to ensure that each time a technical term is mentioned for the first time, a short explanation is included (for example AMR in Line 75 – Line 80; HPCIAs in Line 103 – Line 111. For statistical measures such as confidence interval and odds ratio, we trust that our readers possess familiarity with these concepts, making it sufficient to provide the full terms without further elaboration. In response to your comment, we have decided not to remove the list of abbreviations, as suggested by reviewer 1. We believe that including the list will enhance the article's accessibility for a broader audience.
Comment 4: Reorganize the methodology details, so that the readers could understand effectively. To achieve the goal, you should modify your original text according to the following tips.
- Add a Brief Overview Paragraph:
- At the beginning of the Methods section, include a short overview paragraph summarizing the study’s main stages. This could provide a high-level view of the study design, sample collection, AMU and AMR measurement, and data analysis. For instance:
- Example Overview:
- “This study was conducted on 19 broiler farms in West Java, Indonesia, focusing on the relationship between antimicrobial use (AMU) and antimicrobial resistance (AMR) in coli isolates. Data was collected across multiple production cycles to assess AMU patterns and susceptibility of E. coli isolates to various antimicrobials. Statistical analyses, including Spearman correlation and generalized linear mixed models, were used to evaluate the correlation between AMU and AMR at both farm and antimicrobial class levels.”
- Insert Subheadings for Key Study Steps:
Breaking down the section with clear subheadings would guide readers through the process in an organized way. Suggested subheadings might include:- “Farm and Sample Selection”
- “Data Collection on Antimicrobial Use (AMU)”
- “Isolation and Susceptibility Testing of coli”
- “Data Analysis and Statistical Methods”
- Add a Simplified Flowchart:
- Include a flowchart at the beginning of the section to visually represent the study’s workflow. This could show the progression from farm selection to data analysis in a straightforward, visual format, allowing readers to grasp the methodology quickly before reading the detailed descriptions.
Response 4: We have taken your suggestions into account and restructured the Materials and Methods section of the manuscript based on your recommendation in Line 471 – Line 603. To enable readers to quickly grasp the methodology, we’ve added a flowchart describing the process of the data collection and analysis (Figure 6, Line 478).
Comment 5: Please include a short section in the discussion about the potential policy and global health implications of this study would enhance the manuscript’s impact, especially for readers from different fields.
Response 5: We appreciate your suggestion. In Line 460 to Line 469 we have added a paragraph in the discussion regarding the potential policy implications of our study.
Regarding the global health implications we have added a sentence to make this more apparent in Line 433 – Line 437.
Round 2
Reviewer 2 Report
Comments and Suggestions for Authors
The manuscript has been improved well, therefore i suggest to accept it for publication